# Co-Immunization with DNA Vaccines Expressing SABP1 and SAG1 Proteins Effectively Enhanced Mice Resistance to *Toxoplasma gondii* Acute Infection

**DOI:** 10.3390/vaccines11071190

**Published:** 2023-07-02

**Authors:** Xiaoyu Sang, Xiang Li, Ran Chen, Ying Feng, Ting He, Xiaohan Zhang, Saeed El-Ashram, Ebtsam Al-Olayan, Na Yang

**Affiliations:** 1Key Laboratory of Livestock Infectious Diseases, Shenyang Agricultural University, Ministry of Education, Shenyang 110866, Chinalixiang1998@stu.syau.edu.cn (X.L.); chenran@syau.edu.cn (R.C.); myfengying@syau.edu.cn (Y.F.); 2021240709@stu.syau.edu.cn (T.H.); zhangxiaohan7@stu.syau.edu.cn (X.Z.); 2College of Animal Science and Veterinary Medicine, Shenyang Agricultural University, Shenyang 110866, China; 3Department of Zoology, Faculty of Science, Kafrelsheikh University, Kafr El-Sheikh 33516, Egypt; saeed_elashram@yahoo.com; 4Department of Zoology, College of Science, King Saud University, Riyadh 11451, Saudi Arabia; eolayan@ksu.edu.sa

**Keywords:** *Toxoplasma gondii*, DNA vaccine, SABP1, SAG1, co-immunization

## Abstract

*Toxoplasma gondii* (*T. gondii*) has many intermediate hosts, obligately invades nucleated cells, and seriously threatens human and animal health due to a lack of effective drugs and vaccines. Sialic acid-binding protein 1 (SABP1) is a novel invasion-related protein that, like surface antigen 1 (SAG1), is found on the plasma membrane of *T. gondii*. To investigate the immunogenicity and protective efficacy of DNA vaccines expressing SABP1 and SAG1 proteins against *T. gondii* acute infection, the recombinant plasmids pVAX1-SABP1 and pVAX1-SAG1 were produced and administered intramuscularly in Balb/c mice. Serum antibody levels and subtypes, lymphocyte proliferation, and cytokines were used to assess immunized mice’s humoral and cellular immune responses. Furthermore, the ability of DNA vaccines to protect mice against *T. gondii* RH tachyzoites was tested. Immunized mice exhibited substantially higher IgG levels, with IgG2a titers higher than IgG1. When the immune group mice’s splenocytes were stimulated with *T. gondii* lysate antigen, Th1-type cytokines (IL-12p70, IFN-γ, and IL-2) and Th2-type cytokine (IL-4) increased significantly. The combined DNA vaccine significantly increased the immunized mouse survival compared to the control group, with an average death time extended by 4.33 ± 0.6 days (*p* < 0.0001). These findings show that DNA vaccines based on the SABP1 and SAG1 genes induced robust humoral and cellular immunity in mice, effectively protecting against acute toxoplasmosis and potentially serving as a viable option for vaccination to prevent *T. gondii* infection.

## 1. Introduction

Toxoplasmosis is a zoonotic disease that is found all over the world and poses a serious threat to human and animal health. *Toxoplasma gondii* (*T. gondii*), causing toxoplasmosis, has a diverse host range, ranging from mammals to birds, and actively invades almost all nucleated cells [1]. Mammals (including humans) could become infected with *T. gondii* by consuming contaminated food or water and undercooked meat containing tissue cysts [2,3]. Individuals with complete immunity are usually asymptomatic when infected; however, the infection is generally severe and sometimes fatal in individuals with low or compromised immunity, especially pregnant women. The acute *T. gondii* infection rate of pregnant women during pregnancy is about 0.6%, and the total risk of mother-to-child transmission is about 30%. Most infants have no symptoms of infection at birth, but up to 90% of infants will have sequelae, including choroid, congenital cataracts, and neurocognitive disorders, which have a huge impact on health systems [4]. Toxoplasmosis also causes significant economic losses in the livestock industry, frequently resulting in abortion and stillbirth in pregnant animals, particularly goats and sheep [5]. Some drugs, such as sulfonamides, have some effect on acute toxoplasmosis but almost no impact on chronic toxoplasmosis. Furthermore, the negative effects of these drugs on the human body cannot be overlooked. As a result, research into *T. gondii* vaccines is critical for toxoplasmosis prevention [6,7]. Toxovax, the only approved vaccine for ovine toxoplasmosis, is derived from the non-cyst-forming S48 strain. However, no vaccine protects against *T. gondii* infection in humans or other animal species. Thus, a novel, safe, and effective vaccine to prevent and control toxoplasmosis has always been investigated [8,9]. *T. gondii* tachyzoite actively invades nucleated cells of the host, and surface antigens, micronemal proteins, dense granule proteins, and rhoptry protein family proteins all play important roles in this process. Researchers have investigated various types of vaccines based on invasion and virulence-related proteins, such as SAG1, SAG2, and SAG3; MIC1, MIC2, MIC3, MIC5, MIC6, and MIC8; GRA1, GRA2, GRA4, GRA6, GRA7, GRA24, and GRA25; and ROP2, ROP8, ROP17, ROP18, and ROP35 [10,11,12,13,14,15,16,17,18]. Compared to other vaccines, DNA vaccines offer many benefits of low cost, safety, and ease of preservation, among other things [19]. More importantly, MHC class I and II molecules could present the DNA vaccine, stimulating humoral and cellular immunity. Previous studies demonstrated that multi-gene DNA vaccines’ protective ability was superior to single DNA vaccines due to the complex invasion process of *T. gondii* into the host cell [20]. As a major tachyzoite surface antigen, the SAG1 protein was an ideal vaccine candidate. Thus, several DNA vaccine cocktails combining SAG1 with ROP1, ROP2, ROP18, MIC3, or SAG3 could potentially boost humoral and cellular immunity and mouse survival against *T. gondii* [21,22,23,24,25,26].

Previous studies in our lab found a novel invasion-related protein called sialic acid binding protein-1 (SABP1), which was found on the plasma membrane surface of *T. gondii* [27]. The SABP1 protein belonged to the immune-mapped protein (IMP) family and possessed an IMP2N-terminal domain that was highly conserved in the Apicomplexa, according to phylogenetic analysis. Some members of the IMP family have been used to create DNA vaccines that may increase host resistance to pathogens [28,29,30,31].

In this study, the *SABP1* and *SAG1* genes of *T. gondii* were utilized to develop DNA vaccines, and the effect of single or combined immunization was tested in a Balb/c mouse model. Our findings demonstrated that *SABP1* and *SAG1* gene DNA vaccines induced robust humoral and cellular immunity in mice compared to control groups, effectively enhancing mouse ability to resist acute infection with the *T. gondii* RH strain.

## 2. Materials and Methods

### 2.1. Mice and Parasites

Six-week-old female Balb/c mice were purchased from Liaoning Changsheng Biotechnology Company (Laioning, China). Shenyang Agricultural University’s animal husbandry guidelines were strictly followed during mouse breeding and experimentation. The laboratory animal experiments were approved by Shenyang Agricultural University’s institutional ethics committee (Permit No. 2022111001). Vero cells were used to culture *T. gondii* RH tachyzoites. Tachyzoites were used to prepare *Toxoplasma* lysate antigen (TLA), total RNA extraction, and a mouse challenge experiment.

### 2.2. Polyclonal Antibody Preparation against Recombinant SABP1 and SAG1 Proteins Expressed by E. coil

TgSAG1 primers were designed as forward 5′ GGAATTCCATATGGCGTCGCATTCTC 3′ and reverse 5′ ATAGTTTAGCGGCCGCCTCGAGAGCAG 3′. The *SAG1* gene coding sequence was amplified by PCR from *T. gondii* RH strain cDNA and subcloned to vector pET28a. *E. coli Transtta* (DE3) (TransGen Biotech, Beijing, China) was transformed with the recombinant plasmid pET28a-SAG1. The recombinant SAG1 protein was expressed in *E. coli Transtta* (DE3) and purified using an Ni^2+^–NTA agarose column via affinity chromatography. The expression and purification of recombinant SABP1 protein were carried out as previously described [27]. Purified SABP1 and SAG1 proteins were used to generate polyclonal antibodies. Intravenously immunizing animals with pure SABP1 or SAG1 protein (50 µg/mouse) produced a polyclonal antiserum. Each mouse was given three injections at two-week intervals, and blood was drawn two weeks after the fourth injection [23].

### 2.3. Eukaryotic Expression Plasmid Construction

The primer sequences that were designed are as follows: pVAX1-SABP1-5: 5′ AACTTAAGCTTGCCACCATGGGATCTGGCAACAAC 3′,pVAX1-SABP1-3: 5′ TCCGTCTAGATCAATGGTGATGGTGATGATGCTTC 3′,pVAX1-SAG1-5: 5′ ACCCAAGCTTATGGGCAGCAGCCAT 3′,pVAX1-SAG1-3: 5′ CTAGTCTAGATCAGTGGTGGTGGGTGGGTGGGT 3′.

PCR amplified the *SABP1* and *SAG1* gene coding sequences from prokaryotic expression plasmids and subcolonized them into the pVAX1 plasmid via enzyme digestion and ligation. The pVAX1-SABP1 and pVAX1-SAG1 constructs were then transferred into *E. coli* DH5α (Takara Biomedical Technology, Beijing, China), and positive clones were utilized for DNA sequencing. An Endofree Maxi Plasmid kit was used to purify the plasmids. At OD_260_ and OD_280_, the concentrations of pVAX1-SABP1 and pVAX1-SAG1 were measured using a spectrophotometer.

### 2.4. IFA and Western Blot Detection of pVAX1-SABP1 and pVAX1-SAG1 In Vitro

The HEK293T cells were transfected with pVAX1-SABP1 and pVAX1-SAG1 utilizing the Lipofectamine 6000 reagent per the manufacturer’s directions. Indirect immunofluorescence assay (IFA) and Western blot were used to detect cell recombinant plasmid expression as previously described [15,16,17,18]. In brief, HEK293T cells were fixed for 10 min at RT with 4% paraformaldehyde. After PBS washing, cells were penetrated for 10 min with 0.2% Triton X-100. Anti-*T. gondii* polyclonal antiserum (1:1000 dilution in PBS) was employed as the primary Ab, and Alexa Fluor 488 goat anti-mouse IgG (H + L) (Invitrogen, Shanghai, China) was utilized as the secondary Ab (1:1000 dilution in PBS). The negative control was HEK293T cells transfected with the pVAX1 plasmid. The substance Gold Antifade Mountant with DAPI (Thermo Fisher, Shanghai, China) was dropped with ProLong, which was subsequently examined under the fluorescence microscope. The expression of pVAX1-SABP1 and pVAX1-SAG1 in HEK293T cells was also confirmed using Western blot analysis. Transfected HEK293T cells were resuspended in 20 μL 5 × SDS-PAGE sample buffers. The primary antibody was anti-*T. gondii* polyclonal antiserum, and the secondary Ab was HRP-labeled Goat Anti-Mouse IgG (H + L) (Beyotime Biotechnology, Shanghai, China). An ECL Substrate kit (Qinxiang Scientific Instrument, Shanghai, China) was used to visualize the protein.

### 2.5. Mouse Immunization and T. gondii Challenge

The experimental groups (12 mice/group) consisted of mice immunized intramuscularly four times at 2-week intervals with 100 µL (1 µg/µL) plasmids containing pVAX1-SABP1, pVAX1-SAG1, and pVAX1-SABP1 + pVAX1-SAG1, respectively. Mice were given 100 µL PBS (negative control group), 100 µg empty pVAX1 plasmid (vector control group; 1 µg/µL), and no treatment (blank control). Blood was drawn from the tail veins of each group of mice at 2, 4, 6, and 8 weeks [17]. Sera were collected and kept at −20 °C after being centrifuged for 5 min at 4000× *g*. Mice (6 mice/group) were intra-peritoneally infected with 50 RH strain tachyzoites 2 weeks after the four vaccines, and survival was monitored daily. In addition, six mice were slaughtered from each group to investigate humoral and cellular responses.

### 2.6. Antibody Response Measurement

Anti-*T. gondii* IgG, IgG1, and IgG2a Abs were identified in serum samples using ELISAs as previously described [21,22,23]. In brief, 100 μL (10 μg/mL) TLA diluted in 0.05 M potassium phosphate buffer pH 8 was coated onto microtiter plates and incubated overnight at 4 °C. The plates were washed 3× with PBS containing 0.05% Tween20 (PBST) and blocked at 37 °C for 1 h with PBST having 1% BSA. After washing with PBST, the plates were incubated at RT for 1 h with serum diluted 1:100 in PBS and examined with HRP-conjugated goat anti-mouse IgG1 (ImmunoWay Biotechnology Company, Plano, TX, USA), IgG2a (ImmunoWay Biotechnology Company, Plano, TX, USA), or IgG (ImmunoWay Biotechnology Company, Plano, TX, USA) as secondary Abs for isotype analysis. Each well’s peroxidase activity was measured by adding 100 μL of TMB solution (Beyotime Biotechnology, Shanghai, China). Using an ELISA microplate reader, the OD was measured at 450 nm after halting the reaction with 50 µL of 2 M H_2_SO_4_. Each sample was evaluated in triplicate.

### 2.7. Spleen Lymphocyte Proliferation Test (CCK-8)

Two weeks after the last immunization, splenocyte suspensions from six mice in each group were prepared by forcing the spleens through a wire mesh, purified by removing the RBCs with erythrocyte lysis buffer, and then resuspended in 1640 medium containing 10% FBS. In short, 5 × 10^5^ cells per well were grown in 96-well plates and stimulated for 48 and 72 h at 37 °C with 5% CO_2_ with TLA (20 µg/mL), ConA (positive control; 5 µg/mL) (Solarbio, Beijing, China), GST (20 g/mL), or 1640 medium alone (negative control). Then, 10 µL of CCk8 (Glpbio, Shanghai, China) was added to each well, followed by 4 h incubation. The following formula was utilized to determine the stimulation index (SI): (OD_450_ TLA-OD_450_ Blank): (OD_450_ ConA-OD_450_ Blank) = stimulation index (SI). Three separate mice were utilized to collect and examine the samples [18].

### 2.8. Cytokine Assays

Spleen cells were cultured in 96-well plates for the lymphocyte proliferation assay and induced with TLA (20 µg/mL). At 24 and 96 h, cell-free supernatants were collected and analyzed for IL-2, IL-4, and IFN-γ activity using an ELISA reagent for murine IL-12(p70), IFN-γ, IL-2, and IL-4. The four cytokine concentrations were determined using an ELISA kit (Elabscience, Shanghai, China), and all tests were done in triplicate [21].

### 2.9. Statistical Analysis

Statistical analyses in this experiment were performed utilizing GraphPad Prism 9.0. The normal distribution statistics were presented as mean ± SD. The antibody titer data, lymphoproliferative assay, cytokines, and mean death time were analyzed in univariate variance using Dunnett’s multiple comparisons tests, with *p* < 0.05 considered substantially significant (**** *p* < 0.0001, *** *p* < 0.001, ** *p* < 0.01, and * *p* < 0.05).

## 3. Results

### 3.1. Bioinformatic Analysis Identified B and T Cell Epitopes of the SABP1 and SAG1 Proteins

Both the SABP1 and SAG1 proteins were detected on the tachyzoite surface and engaged in the parasite’s host cell invasion [27]. Because of its immunogenicity, the SAG1 protein was commonly used as a reference. The antigenic analysis of the SABP1 protein was compared to the SAG1 protein. By DNASTAR analysis, the SABP1 protein had a higher score for hydrophilicity, flexible region, surface probability, and antigenic index than the SAG1 protein, as shown in Figure 1A,B. Furthermore, the IEDB online service examined the T cell epitopes of the SABP1 and SAG1 proteins. MHC class II molecules were linked to IC_50_ values from the SABP1 and SAG1 peptides, whose minimum percentile ranks are shown in Table 1, indicating that the two membrane proteins likely had good binding to MHC-II.

### 3.2. Preparation of Polyclonal Antibody of Recombinant SABP1 and SAG1 Proteins Expressed by E. coli

The *SABP1* and *SAG1* gene coding sequences were amplified by PCR from *T. gondii* cDNA and cloned into the prokaryotic expression vectors pDEST17 and pET28a, respectively, to generate pDEST17-SABP1 and pET28a-SAG1 [27]. We then transferred them into *E. coli* BL21 (DE3) and *E. coil Transetta* (DE3). After induction and purification, SDS-PAGE analysis revealed highly pure recombinant SABP1 and SAG1 proteins (Figure 2A,B). Immunizing mice produced polyclonal antibodies to recombinant SABP1 and SAG1 proteins, and they specifically recognized the native proteins in a *T. gondii* lysate antigen (TLA) preparation (Figure 2C,D).

### 3.3. Expression of pVAX1-SABP1 and pVAX1–SAG1 in HEK293T Cells

The constructed pVAX1-SABP1 and pVAX1-SAG1 eukaryotic expression vectors were transfected into HEK293T cells. Using indirect immunofluorescence, green fluorescence was respectively observed in cells transfected with pVAX1-SABP1 and pVAX1-SAG1, while no green fluorescence was seen in the blank cells or pVAX1-transfected cells (Figure 3A). Simultaneously, lysed cells were detected by western blot, with positive bands found in cells transfected with pVAX1-SABP1 and pVAX1-SAG1. There was no band in the blank or pVAX1-transfected cells (Figure 3B,C). These findings suggested that SABP1 and SAG1 proteins might be receptively produced in eukaryotic cells using pVAX1 vectors.

### 3.4. Combined DNA Vaccines Induced Mice to Produce Higher Levels of IgG and Subtype IgG1, IgG2a Abs

IgG1 and IgG2a isotype distribution was tested two weeks after the final inoculation, and the total IgG antibody titer was used to identify antibody levels triggered by four doses of a single or combined vaccine. *T. gondii* lysate antigen (TLA), used as a coated antigen for the ELISA test, was specifically recognized by the serum isolated from the mice immunized with pVAX1-SABP1 and pVAX1-SAG1 (Figure 4A). The antibody levels of the immunized mice gradually increased with the lengthening of immunization times, but not those of the negative and blank groups (Figure 4B). The combined immunization group’s mice had significantly higher antibody levels than the other mice (*p* < 0.05). The IgG2a level was higher than IgG1 in all vaccination groups, indicating that the DNA vaccine elicited more potent Th1-type cellular immunity in the mice (Figure 4C).

### 3.5. Splenocyte Proliferation

The spleen is a vital peripheral immune organ that contains many immune cells. CCK8 was used to examine the splenocyte proliferative levels induced by TLA, ConA (positive stimulation), GST (negative stimulation), or 1640 (blank). Figure 5 depicts the proliferation stimulation index (SI). Significant splenocyte proliferation was observed only after ConA stimulation in the three control groups (blank, PBS, and pVAX1). However, in groups immunized with single or combined DNA vaccines, the proliferation stimulation index induced by TLA and ConA was significantly higher than in GST or 1640 simulations.

### 3.6. Th1 Cytokines (IFN-γ, IL-12p70, IL-2) and Th2 Cytokines (IL-4) Significantly Increased Levels in Combined Immunized Mice

We utilized an ELISA method to detect Th1- and Th2-type cytokines released in culture supernatants after TLA induction of mouse spleen cells to determine the T helper cell response type. As illustrated in Figure 6, spleen cells of mice immunized with a combined DNA vaccine secreted significantly more IL-12p70, IFN-γ, IL-2, and IL-4 than the other groups. Furthermore, four cytokines were substantially higher in the pVAX1-SABP1 group than in the pVAX1-SAG1 group, specifically IFN-γ and IL-12p70. However, after exposure to TLA, the spleen cells of the blank, PBS, and pVAX1 groups produced very few cytokines.

### 3.7. Combined DNA Vaccines Immunization Effectively Prolonged the Survival Time of T. gondii-Infected Mice

All experimental mice (*n* = 6) were challenged with 50 tachyzoites (*T. gondii* RH strain) to examine the protection given by single or combination DNA vaccinations. As illustrated in Figure 7, all mice in the blank, PBS, and pVAX1 groups died 9 days after challenges. However, mice died on the tenth day after the challenge in the immunized group with combined DNA vaccines, and the longest survival time was 14 days. The combined immunization group’s average death time was significantly longer than the blank, PBS, and PVAX1 groups (*p* < 0.001).

## 4. Discussion

In recent years, the research and development of new vaccines, such as the nanoparticle vaccine, the virus-like particle vaccine, and the live attenuated vaccine, provide new strategies for the prevention and control of human toxoplasmosis [6,7]. With continued multi-disciplinary efforts, successfully developing a toxoplasmosis vaccine for humans may be promising. Because *T. gondii* obligately parasitizes in host nucleated cells, both humoral and cellular immunity are required for host resistance to infection. Furthermore, cellular immunity is important in the host’s clearance of intracellular *T. gondii* [32]. A previous study has compared the immunoprotective efficacy of the SAG1 protein to that of other virulence-related proteins, such as ROP1, ROP2, ROP18, MIC3, and SAG3 [21,22,23,24,25,26]. Immune-mapped protein 1, which possessed an IMP2N-terminal domain, was found in *Eimeria maxima* and shown to confer immunity against *E. maxima* [28]. Jia et al. found IMP1 homologs in all apicomplexan parasites and classified them into six groups using a phylogenetic tree based on sequence, with TgIMP1 falling into IMP1 groups and TgSABP1 falling into IMP4 groups [31]. Recent research demonstrates that immunizing Balb/c mice with the TgIMP1 DNA vaccine increased the survival time of mice challenged with the lethal *T. gondii* RH strain tachyzoites compared to the control group [33,34,35]. In this study, DNA immunization with pVAX1-SAG1 and/or pVAX1-SABP1 induced humoral and Th1-type cellular immunity, which may have contributed to the enhanced survival time of mice challenged with the *T. gondii* RH strain. In line with previous research, these two antigen-based DNA cocktail vaccines induced stronger protective immunity than single antigen-based DNA immunization [36,37].

Ab-mediated humoral immunity is vital in preventing the proliferation of *T. gondii* tachyzoites [38]. The SABP1 and SAG1 proteins were chosen to construct DNA vaccines in this study because previous research showed that they were both located on the *T. gondii* tachyzoite plasma membrane and that their antibodies could significantly block the parasite’s invasion into host cells in vitro [27]. Anti-*T. gondii* IgG levels were substantially higher in mice immunized with double DNA vaccines than in the others, which could help limit *T. gondii* infection (Figure 2C) [39,40]. Furthermore, compared to the control groups, DNA immunization with a single DNA vaccine, particularly a combined DNA vaccine, significantly increased IgG2a and IgG1 levels. Meanwhile, the higher IgG2a/IgG1 ratio in immunized groups indicates that Th1 immunity predominated [41,42,43]. Cellular immunity is critical for the clearance of intracellular *T. gondii*. The spleen, as an important peripheral immune organ, is home to many lymphocytes and other immune cells, and the antigen could cause immune cells in the spleen to become activated and proliferate [44]. The stimulation of ConA and TLA, but not GST, significantly increased splenocyte proliferation in all immunized groups. However, only ConA stimulation increased splenocyte proliferation in the control and blank groups. These findings indicated that the immunized groups had effective cellular immunity against *T. gondii* infection.

T helper type 1 (Th1) cells generate many cytokines, such as IFN-γ, IL-12, and IL-2, and protect the host from *T. gondii* infection [45]. IFN-γ is the major cytokine for enabling host resistance to *T. gondii* infection through several protective mechanisms, including inducing macrophage cells towards M1 polarization, producing nitrogen oxides (NO) to clear *T. gondii* by induced tryptophan degradation, and recruiting immune-related GTPases (IRGs) to attack parasitophorous vacuoles [45,46,47,48,49]. Interleukin 2 (IL-2) is a major cytokine produced by Th1 cells that promotes T cell proliferation and differentiation while also increasing the activation of CTL and NK cells, vital for resistance to *T. gondii* infection [50]. Interleukin 12 (IL-12), which is required to produce IFN-γ, promotes Th0 to Th1 cell transformation. Because of a weakened host Th1 immune response, blocking or inhibiting IL-12 expression contributed to *T. gondii* growth and dissemination in both the acute and chronic phases of infection [51]. Interleukin 4 (IL-4), as Th2-cell-biased cytokines, promotes B cell proliferation and antibody production, contributing to humoral immunity against *T. gondii* infection [45]. In the current study, significantly higher levels of IL-12p70, IFN-γ, IL-2, and IL-4 were identified in mice immunized with single or double DNA vaccines, indicating that DNA vaccines successfully elicited Th1- and Th2-type mediated immunity in mice. However, the SABP1 DNA vaccine alone induced the immune cells of the host to produce significantly more cytokines than the SAG1 DNA vaccine dose, which may be related to the expression efficiency of these two genes in mammal cells (Figure 3 and Figure 6). Furthermore, IgG2a levels were higher than IgG1 in all immunized groups, indicating that Th1-type immune responses were predominant.

Our findings show that using *SABP1* and *SAG1* genes in a multivalent DNA vaccine induces stronger immune responses and protective effects than a single-gene vaccine. When mice were challenged with *T. gondii* lethal doses, all controls were immunized with PBS, and the empty plasmid group died within 7–9 days. The single-gene immunization group showed low protection against the *T. gondii* RH strain challenge, whereas combined DNA vaccine immunization significantly extended mouse survival time, with the average time of death extended by 4.33 ± 0.6 days.

## 5. Conclusions

The current study shows that the combined DNA vaccines pVAX1-SABP1 and pVAX1-SAG1 are the potential for DNA vaccines that elicit immune protection against acute *T. gondii* infection in mice. Notably, pVAX1-SABP1 was more readily expressed in mammalian cells than pVAX1-SAG1 and induced mice to produce a stronger cellular response. Thus, more research is required to examine the protective immunity stimulated by utilizing the *SABP1* gene with more *T. gondii* antigens.

## Figures and Tables

**Figure 1 vaccines-11-01190-f001:**
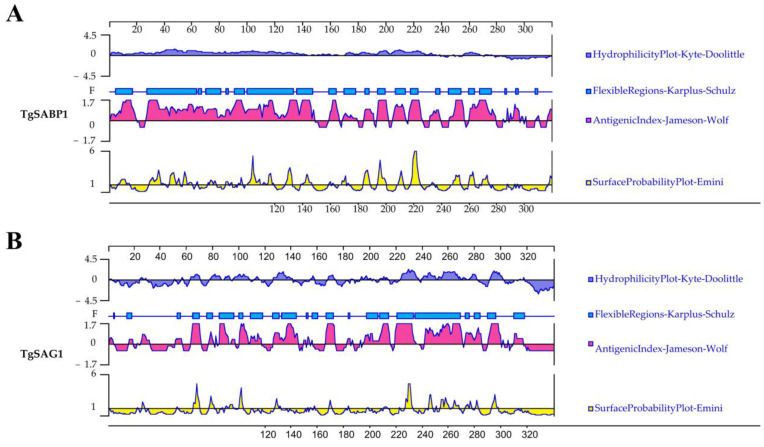
Antigenic characteristics of SABP1 and SAG1 proteins analyzed by bioinformatics. The antigen index, hydrophilicity, flexibility, and surface probability of SABP1 and SAG1 proteins predicted by DNASTAR (**A**,**B**).

**Figure 2 vaccines-11-01190-f002:**
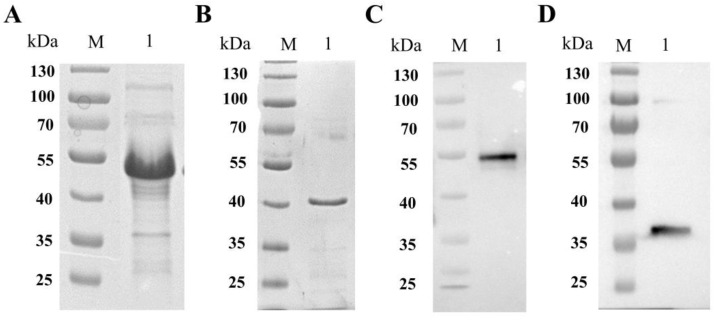
Expression and purification of SABP1 and SAG1 recombinant proteins and specificity analysis of their polyclonal Abs produced from immunized mice. (**A**) Purified SABP1 recombinant protein by SDS-PAGE; (**B**) Purified SAG1 recombinant protein by SDS-PAGE; (**C**) TLA specifically recognized by the polyclonal Ab of recombinant SABP1 protein; (**D**) TLA specifically recognized by the polyclonal antibody of recombinant SAG1 protein.

**Figure 3 vaccines-11-01190-f003:**
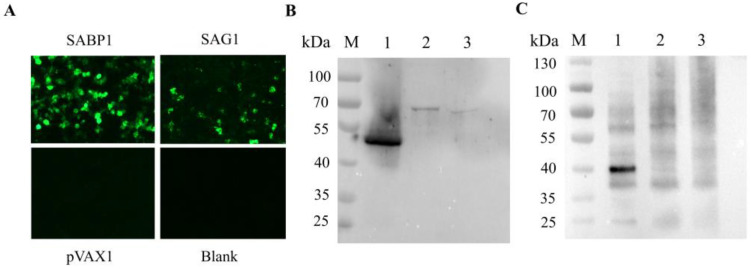
Expression of pVAX1-SABP1 and pVAX1-SAG1 in HEK293T cells. (**A**) The expression of pVAX1-SABP1, pVAX1-SAG1, and pVAX1 in HEK293T cells detected by an indirect immunofluorescence assay; (**B**) The expression of pVAX1-SABP1 in HEK293T cells identified by western blot. 1: pVAX1-SABP1 transfected cells; 2: pVAX1 transfected cells; 3: non-transfected cells; (**C**) The expression of pVAX1-SAG1 in HEK293T cells identified by western blot. 1: pVAX1-SAG1 transfected cells; 2: pVAX1 transfected cells; 3: non-transfected cells.

**Figure 4 vaccines-11-01190-f004:**
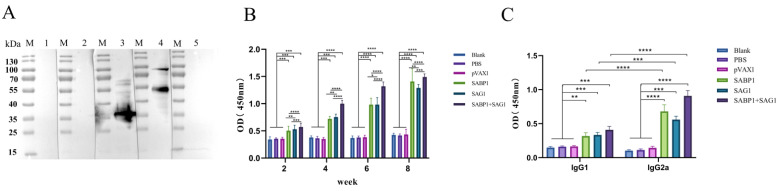
Detection of the total IgG, IgG1, and IgG2a Ab levels in the immunized mouse sera. (**A**) TLA specially recognized by polyclonal Ab in the mouse immunized serum through western blot; 1: Blank; 2: PBS; 3: pVAX1-SAG1; 4: pVAX1-SABP1; 5: pVAX1 vector; (**B**) IgG Ab determination in the sera of Balb/c mice at 2, 4, 6, and 8 weeks post the first vaccination; (**C**) IgG1 and IgG2a Ab detection in immunized mice 14 days after the last immunization. Data are displayed as mean ± SD (*n* = 6) in every group (* *p* < 0.05, ** *p* < 0.01, *** *p* < 0.001, **** *p* < 0.0001).

**Figure 5 vaccines-11-01190-f005:**
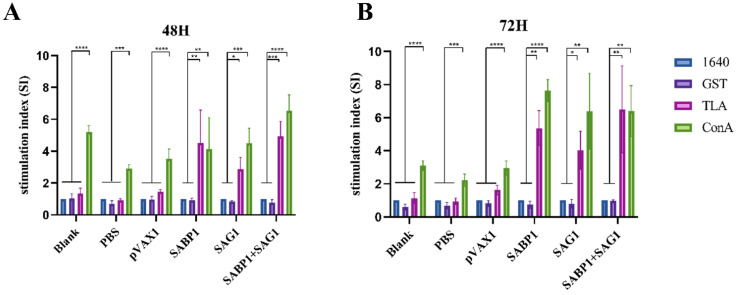
Spleen cell proliferation of mice induced by different stimulants 14 days after the fourth immunization. (**A**) Proliferation of spleen cells after 48 h of stimulation; (**B**) Proliferation of spleen cells after 72 h of stimulation. Lymphocyte proliferation stimulation index (SI). Data are displayed as mean ± SD (*n* = 6) in every group (* *p* < 0.05, ** *p* < 0.01, *** *p* < 0.001, **** *p* < 0.0001).

**Figure 6 vaccines-11-01190-f006:**
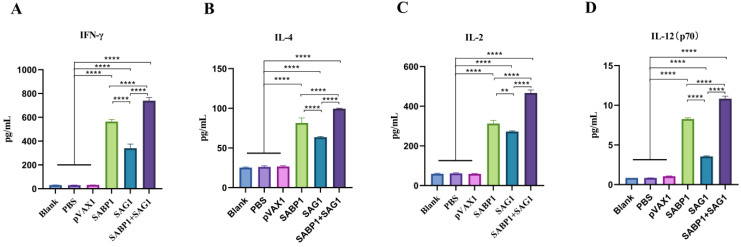
TLA-stimulated mouse splenocyte cytokine levels. Production of IFN-γ (**A**), IL-4 (**B**), IL-2 (**C**) and IL-12 (p70) (**D**) in the cell supernatant of splenocyte in all experimental groups mice. Data are displayed as mean ± SD (*n* = 6) in every group (** *p* < 0.01, **** *p* < 0.0001).

**Figure 7 vaccines-11-01190-f007:**
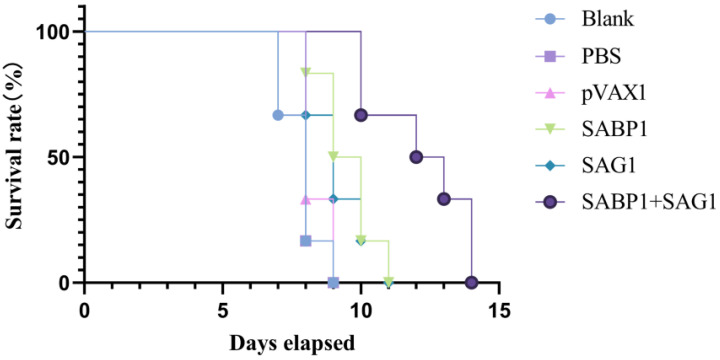
Balb/c mice challenged with 50 RH tachyzoites survived 14 days following the previous inoculation.

**Table 1 vaccines-11-01190-t001:** IC_50_ values for SABP1 and SAG1 binding to MHC class II molecules utilizing IEDB.

MHC II Allele ^1^	Start-Stop ^2^		Percentile Rank ^3^	
	SAG1	SABP1	SAG1	SABP1
H2-IAb	26–40	33–47	0.95	1.7
H2-IAd	20–34	20–34	2.75	9.55
H2-IEd	14–28	10–24	3.35	0.25

^1^ H2-IAb, H2-IAd, and H2-IEd alleles are mouse MHC class II molecules; ^2^ 15 amino acids were chosen for analysis; ^3^ Low percentile denotes high-level binding.

## Data Availability

The datasets generated during the current study are available from the corresponding author on reasonable request.

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
