# Peer review of "Co-Immunization with DNA Vaccines Expressing SABP1 and SAG1 Proteins Effectively Enhanced Mice Resistance to Toxoplasma gondii Acute Infection"

_vaccines, 2023, doi:10.3390/vaccines11071190_

Round 1
Reviewer 1 Report
In the current manuscript, Sang et al. investigated the efficacy of co-immunization with DNA vaccines expressing SABP1 and SAG1 proteins in enhancing resistance of mice to Toxoplasma gondii infection. Specifically, they found that co-immunization with these two proteins by DNA vaccines can enhance both humoral and cellular immune response against T. gondii and extend the lives of mice challenged with acute infection of this pathogen. Overall, the manuscript is fairly well-written, and the results are somewhat promising. However, there are still some major inconsistencies and omissions that must be addressed:
1. In the Materials and Methods, they wrote “Six-week-old female Balb/c mice were purchased…” Why did they only use female mice for this study?
2. In Fig 2, the recombinant SAPB1 is around 55kDa (panel A) and the recombinant SAG1 protein is around 40kDa (panel B). However, in panel D where they detect SAG1 protein with the polyclonal Ab, the size of SAG1 protein detected is way below 40kDa. Similarly, In Fig 3 B, the size of SABP1 protein is now way above 55 kDa and in Fig 4 B, the size of SAG1 protein is around 35 kDa. Please explain these discrepancies.
3. Moreover, in Fig 2 C and D, where they detected SABP1 and SAG1 protein by polyclonal Ab from immunized mice, they did not include results from polyclonal Ab of control mice immunized with Blank, PBS, or empty pVAX1 vector so show the specificity of their immunization strategies.
4. Why did they not measure the cytokines (INF-y, IL-2, IL-4, etc.) directly in the serum of the immunized and control mice challenged with T. gondii (in Fig. 7)?
5. In Fig 5, they evaluate the proliferative response of splenocytes isolated from immunized mice by different stimulants. They need to further determine of the percentages of CD4+ or CD8+ T cells in immunized and control mice.
The quality of English language is acceptable, however, there are still typos and grammatical errors throughout the manuscript. The authors need to carefully proof-read their paper before re-submission.
Author Response
- In the Materials and Methods, they wrote “Six-week-old female Balb/c mice were purchased…” Why did they only use female mice for this study?
Response: Thanks for your questions and suggestions. Firstly, previous study demonstrated that the estrogen produced by female mice could increase their immune response (DOI: 10.1016/j.cellimm.2015.01.018). Meanwhile, female mice are more gentle and easy to operate. So we selected female mice to evaluate the immunize effect of vaccines.
- In Fig 2, the recombinant SAPB1 is around 55kDa (panel A) and the recombinant SAG1 protein is around 40kDa (panel B). However, in panel D where they detect SAG1 protein with the polyclonal Ab, the size of SAG1 protein detected is way below 40kDa. Similarly, In Fig 3 B, the size of SABP1 protein is now way above 55 kDa and in Fig 4 B, the size of SAG1 protein is around 35 kDa. Please explain these discrepancies.
Response: Thanks for your questions and suggestions.
For difference of recombinant SABP1 protein in Figure.2A and Figure.3B, the recombinant SABP1 in Figure.2A is right, which is consistent with our previous study (DOI: 10.1093/infdis/jiaa072). However, due to our careless, the molecular weights presented by protein marker bands in Figure.3 were wrongly labeled. Thermo Scientific Prestained Protein Ladder (#26616) was used to detect the molecular weight of expressed protein. As shown followed figure, the red band is about 70 KDa. In the text, the band of protein markers were wrongly labeled. We have corrected it and changed the right figure in the revised mauscript.
Figure SDS-PAGE band profile of the PageRuler Prestained Protein Ladder (#26616)
For the SAG1 protein, Figure2 B showed the molecular weight of recombinant SAG1, which was expressed by recombinant pET-28a vector in E.coil and possessed 36 more amino acids than native SAG1 protein of T. gondii. So the band of recombinant SAG1 protein is slightly above 40 KDa (Figure 2B), however, the band of native SAG1 protein in TLA was below 40 KDa (Figure.2D and Figure. 4B).
- Moreover, in Fig 2 C and D, where they detected SABP1 and SAG1 protein by polyclonal Ab from immunized mice, they did not include results from polyclonal Ab of control mice immunized with Blank, PBS, or empty pVAX1 vector so show the specificity of their immunization strategies.
Response: Thanks for your questions and suggestion. As shown in Figure 2 C and D, the native SABP1 and SAG1 proteins in TLAs were specially detected by polyclonal antibody from immunized mice with respectively recombinant SABP1 and SAG1 proteins acquired by E.coil. These results present the specificity of two antibodies,which could be used to detect the expression of pVAX1-SABP1 and pVAX1-SAG1 in 293T cells. According to the suggestion of reviewer, we have detected the specificity of polyclonal antibody of control mice immunized with Blank, PBS, or empty pVAX1 vector and placed the results in Figure S4 in the supplement materials.
4.Why did they not measure the cytokines (INF-y, IL-2, IL-4, etc.) directly in the serum of the immunized and control mice challenged with T. gondii (in Fig. 7)?
Response: Thanks for your questions and suggestion. Referring to recently published relevant researches, many studies chose to detect the cytokines secreted by spleen cells stimulated by TLAs (DOI:10.3389/fcimb.2021.686004, DOI:10.1016/j.vetpar.2011.12.007 and DOI: 10.1016/j.vaccine.2010.11.012). Because the half-life of different cytokines were different and the content of cytokines in serum may be lower than the detection line of the reagent kit, we chose to detect the cytokines secreted by spleen cells stimulated by TLAs, to a certain extent, which could reflect the response of immunized mice to T. gondii challenge.
5.In Fig 5, they evaluate the proliferative response of splenocytes isolated from immunized mice by different stimulants. They need to further determine of the percentages of CD4+ or CD8+ T cells in immunized and control mice.
Response: Thanks for your questions and suggestion. According to the reviewer’s suggestion, we did the experiment of classified the T cell subpopulations of immunized and control mice. Unfortunately, there was no significant difference in the CD4 T cell and CD8 T cell subpopulations between the immunized and control mice, so this part of the results was not included in the article. The similar result also existed in some previous studies (DOI: 10.1007/s11686-021-00415-2). We speculated that the immunized mouse’s lymphocytes had transformed into memory T cells, which need specific antigen to stimulate their rapid proliferation and differentiation, such as the splenocyte proliferation test. In the following research, we will pay attention to this issue and improve our experimental plan.
Comments on the Quality of English Language
The quality of English language is acceptable, however, there are still typos and grammatical errors throughout the manuscript. The authors need to carefully proof-read their paper before re-submission.
Response: Thank you very much!
Reviewer 2 Report
Authors present an interesting study investigating the efficacy of a vaccine against Toxoplasma Gondii infection in an animal model. The Methods are well described, and the Results clearly explained. However, Authors missed to include in the Introduction that Toxoplasmosis in not only a danger for livestock but is also a burden for humans, in particular pregnant women, since the transmission rate to the fetus is about 30% PMID: 31404785. Toxoplasmosis in non-immune pregnant women is associated with miscarriage and congenital anomalies for the newborns, among which congenital cataract and neurocognitive disorders (PMID: 36365029, PMID: 34458947), with a huge impact on health systems. I suggest including a paragraph in the Introduction explaining the risks of Toxoplasmosis in human pregnancy and also a paragraph in the Discussion section speculating whether a future vaccine for humans against Toxoplasma could be useful to reduce this burden, as it has been for Rubella vaccine.
Author Response
Authors present an interesting study investigating the efficacy of a vaccine against Toxoplasma gondii infection in an animal model. The Methods are well described, and the Results clearly explained. However, Authors missed to include in the Introduction that Toxoplasmosis in not only a danger for livestock but is also a burden for humans, in particular pregnant women, since the transmission rate to the fetus is about 30% PMID: 31404785. Toxoplasmosis in non-immune pregnant women is associated with miscarriage and congenital anomalies for the newborns, among which congenital cataract and neurocognitive disorders (PMID: 36365029, PMID: 34458947), with a huge impact on health systems. I suggest including a paragraph in the Introduction explaining the risks of Toxoplasmosis in human pregnancy and also a paragraph in the Discussion section speculating whether a future vaccine for humans against Toxoplasma could be useful to reduce this burden, as it has been for Rubella vaccine.
Response: Thanks for your good suggestions. We have added the relevant contents on the risk of T. gondii infection in pregnant women and the future T. gondii vaccine in the introduction and discussion, in the revised manuscript.